# Safety and Effectiveness of Unilateral Transcranial Magnetic Resonance-Guided Focused Ultrasound in Essential Tremor: One-Year Single-Center Real-World Results

**DOI:** 10.3390/neurolint17080131

**Published:** 2025-08-21

**Authors:** Salvatore Iacono, Cesare Gagliardo, Domenico Gerardo Iacopino, Giuseppe Schirò, Rosario Maugeri, Sergio Mastrilli, Valentina Picciolo, Eleonora Bruno, Maurizio Marrale, Massimo Midiri, Marco D’Amelio

**Affiliations:** 1Department of Biomedicine, Neurosciences and Advanced Diagnostics (BIND), University of Palermo, 90127 Palermo, Italy; salvatore.iacono02@unipa.it (S.I.); cesare.gagliardo@unipa.it (C.G.); giuseppeschiro1994@gmail.com (G.S.); valentinapicciolo.vp@gmail.com (V.P.); massimo.midiri@unipa.it (M.M.); 2Neurology Unit, Fondazione Istituto G. Giglio, Cefalù, 90015 Palermo, Italy; 3Neuroradiology Unit, University-Hospital Paolo Giaccone, 90127 Palermo, Italy; eleonora.bruno@policlinico.pa.it; 4Neurosurgical Clinic, AOUP “Paolo Giaccone”, Post Graduate Residency Program in Neurologic Surgery, Department of Biomedicine Neurosciences and Advanced Diagnostics, School of Medicine, University of Palermo, 90127 Palermo, Italy; gerardo.iacopino@unipa.it (D.G.I.); rosario.maugeri@unipa.it (R.M.); 5Neurology Unit, Ospedale Buccheri La Ferla, 90123 Palermo, Italy; segiomastro@gmail.com; 6Department of Physics and Chemistry “Emilio Segrè”, University of Palermo, 90133 Palermo, Italy; maurizio.marrale@unipa.it

**Keywords:** essential tremor, activities of daily living, disability, quality of life, tcMRgFUS, effectiveness, safety

## Abstract

**Background/Objectives**: Essential tremor (ET) is the most common movement disorder worldwide. It negatively affects patients’ activities of daily living (ADL) and quality of life. Unilateral transcranial magnetic resonance-guided focused ultrasound (tcMRgFUS) thalamotomy has been proven as a highly effective and safe treatment option for patients with refractory ET. The aims of this study are to explore the effectiveness and safety of tcMRgFUS thalamotomy in patients with ET in a real-world setting. **Methods**: Patients who underwent tcMRgFUS thalamotomy at the University Hospital of Palermo were prospectively enrolled. Scores obtained by Quality of Life in Essential Tremor Questionnaire (QUEST) and The Essential Tremor Rating Assessment Scale (TETRAS) were compared before and after tcMRgFUS thalamotomy. Predictors of tcMRgFUS thalamotomy effectiveness were explored by multivariable Cox regression analyses. All the adverse events (AEs) during and after the procedure were collected. **Results**: Fifty patients were included (80% male; median age at tcMRgFUS 67.4 years). After procedure, the QUEST score decreased by 46.2%, while TETRAS-ADL and TETRAS Performance (TETRAS-PE) decreased by 52.2% and 51.8%, respectively. Temperature peak and longitudinal lesion diameter positively correlated with the magnitude of QUEST and TETRAS-PE reduction. A higher baseline TETRAS-PE score predicted a good prognosis (HR = HR 6.6 [95% CI: 2.1–21.3]; *p* = 0.001). AEs were mild to moderate and transient, while permanent AE was observed only in one case. **Conclusions**: This real-world study confirms the higher effectiveness and the favorable safety profile of tcMRgFUS thalamotomy in patients with ET by reducing the tremor-related interference in quality of life, disability in ADL, and tremor severity.

## 1. Introduction

Essential tremor (ET) is the most common tremor disorder, with a prevalence of 1.33% worldwide, which is increasing with age [1]. ET is characterized by a 4–8 Hz kinetic tremor, although it often has a postural component as well [1,2]. ET commonly involves the upper limbs (95%), head (41%), voice (29%), and, more rarely, legs (17%) [3]. The etiology of ET is unclear, probably resulting by a genetic and environmental interplay. Indeed, the risk of developing ET is higher in first-degree relatives of patients with ET [4,5]. Whatever the etiology, the pathological mechanism underlying ET is the abnormal functioning of the cerebello-thalamo-cortical pathway. Several pathogenetic hypotheses have been proposed, including (1) the neurodegeneration of Purkinje cells; (2) the central oscillatory hypothesis; and (3) the dysfunction of GABAergic transmission throughout the cerebello-thalamo-cortical pathway [6]. However, ET causes significant disability in patients’ activities of daily living (ADL), also affecting their quality of life [7]. Particularly, a previous study showed that the most commonly affected ADL tasks were drinking (74%) and writing (68%), followed by pouring and others [8]. Thus, daily tasks requiring a high-level of hand dexterity may be more affected by tremor. Higher tremor severity may lead to higher disability in ADL and lower quality of life, subsequently. This might account for the association between psychiatric comorbidities such as anxiety, depression, and ET [9]. To date, the therapeutic options of ET are limited and include propranolol, primidone, and topiramate, as well as any drugs able to increase GABAergic transmission, such as gabapentin and benzodiazepines [10]. However, pharmacological treatments are often ineffective and/or poorly tolerated, burdened by a wide range of adverse events (AEs). In the case of refractory ET, radiofrequency-ablation-mediated thalamotomy and deep brain stimulation of ventral intermediate (VIM) thalamic nucleus have been employed. These are invasive interventions carrying a considerable risk of infections of the central nervous system and intracerebral hemorrhages [11]. Recently, unilateral transcranial magnetic resonance-guided focused ultrasound (tcMRgFUS) lesioning the VIM has attracted increasing interest as a highly effective and non-invasive treatment option for ET [12,13]. To date, the effectiveness of tcMRgFUS in patients with ET has been explored by pilot studies, case series, and clinical trials, but real-word studies are limited worldwide.

In this prospective, real-word cohort study, we explored the safety and effectiveness of tcMRgFUS thalamotomy in patients with drug refractory ET attending to the Movement Disorder Outpatient Service of the University Hospital Paolo Giaccone, Palermo, Italy. Specifically, the aims of this study were to (1) explore the impact of unilateral tcMRgFUS thalamotomy in the perceived tremor-related interference in quality of life and tremor-related ADL disability in a cohort of patients with drug-refractory ET; (2) explore the factors associated with tcMRgFUS thalamotomy’s effectiveness; (3) explore the relationship between tcMRgFUS operating parameters, tremor-related quality of life, and tremor-related disability; and (4) assess the safety of unilateral tcMRgFUS thalamotomy.

## 2. Materials and Methods

### 2.1. Study Design and Participants

A prospective cohort study was carried out by enrolling patients with drug-refractory ET who underwent tcMRgFUS thalamotomy at the University Hospital Policlinico Paolo Giaccone, Palermo. The study inclusion criteria were to have (1) a diagnosis of ET according to the Consensus Statement on the Classification of tremors [2]; (2) a drug-refractory ET; (3) a skull deviation ratio (SDR) ≥ 0.3; (4) availability of complete clinical and tcMRgFUS treatment data; and (5) a post-treatment follow-up of at least one year. No other study inclusion criteria, such as age or medical history, were chosen. Patients not meeting the above inclusion criteria were excluded from the final analysis.

### 2.2. Study Outcomes

Tremor disability and tremor severity were evaluated through the administration of Quality of Life in Essential Tremor Questionnaire (QUEST) and The Essential Tremor Rating Assessment Scale (TETRAS) [14,15]. The QUEST score is a 30-item composite scale developed to measure the impact of ET in quality of life by exploring the perceived tremor-related interference in five domains (communication, work and finance, hobbies and leisure, physical, and psychosocial) [16]. The grade of tremor-related interference in each domain is explored by providing specific questions, which are scored from 0 (no interference) to 4 (always). The sum of each subscore yields the total QUEST score; therefore, higher values indicate higher perceived tremor-related interference in quality of life [16]. TETRAS includes two subscales: TETRAS Activities of Daily Living (TETRAS-ADL), which reflects tremor-related disability, and TETRAS performance (TETRAS-PE), which estimates tremor severity. TETRAS-PE represents an objective measure of the tremor during specific tasks. The maximum TETRAS-PE score is 64, with higher scores indicating higher tremor severity [15]. Tremor assessments were performed at each side by a movement disorder-specialized neurologist who evaluated each participant at baseline and after tcMRgFUS procedure. Also, at each study visit, the participants were invited to fulfill a study-specific questionnaire, which included the following points: self-perceived health status (from 0 to 100), self-perceived quality of life (from 0 to 100), and number of hours of tremor per day (from 0 to 24). According to the TETRAS-PE score variation after tcMRgFUS, the patients were categorized as full responders in the case of a variation of at least 50% after thalamotomy or partial responders when the variation was less than 50%. The TcMRgFUS-operating parameters, including temperature peak, mean provided temperature, SDR, number of provided sonication, mean provided power, and power peak, were also collected.

### 2.3. Adverse Events

An adverse event (AE) was any side effect that occurred during and following tcMRgFUS, directly or indirectly. The AEs’ etiology was categorized as Magnetic Resonance Imaging (MRI) and focused ultrasound-related (i.e., related to the procedure, including water filled helmet, sonications, etc.) and thalamotomy-related (i.e., due to anatomical lesion, edema, etc.). Moreover, the AEs’ severity was established according to the Common Terminology Criteria for AEs’ five categories: (1) mild, asymptomatic, it does not require medical therapy or any medical procedure; (2) moderate, few symptoms, it requires a non-invasive medical treatment; (3) severe, it is a non-life-threatening condition, although it requires specific treatment and may require or prolong hospitalization; (4) life-threatening conditions requiring intensive care unit admission; and (5) death related to AE [17].

### 2.4. TcMRgFUS Thalamotomy Procedure

The procedures were performed using a focused ultrasound (FUS) equipment (ExAblate 4000; InSightec Ltd., Haifa, Israel) that consists of a hemispheric 1024-element phased-array transducer operating at 650 kHz. In our center, this FUS system integrates with an MRI unit operating at 1.5T (Signa HDxt; GE Medical Systems, Milwaukee, WI, USA). The specific details on how the patients were managed for the procedures, including their scalp shaving, stereotactic frame fixation, sealant membrane with the embedded dedicated coil positioning, thalamotomy procedure description, and discharge, were previously described [18,19,20,21]. The procedures were carried out by a dedicated neuroradiologist [C.G] as part of a multidisciplinary team including neurologists [M.D.], neurosurgeons [DG.I.; R.M], MRI technicians, and nurses ensuring comprehensive patient management. The initial target position was determined on MR images using standard stereotactic coordinates: 75% along the anterior commissure–posterior commissure (AC-PC) line, 12–14 mm laterally from the median plane and 0–2 mm caudocranially from the intercommissural plane. The final decision on the treatment target coordinates was also based on the results of a patient-specific probabilistic tractography analysis with thalamic parcellation and target adjustments based on patient feedback during low-power sonications [22,23]. For each patient, treatment planning involved co-registration of CT and MRI scans, with calcifications and other critical structures designated as “no-pass” zones to avoid interference with the high-intensity focused ultrasound (HI-FU) beam path. The number of transducer elements required (minimum of 700) and the usable head surface area (at least 250 cm^2^) were calculated to ensure adequate energy delivery. Fiducial markers were placed on real-time MR images to enable automatic motion tracking. Prior to the procedure, a tracking scan was performed to confirm and register the transducer’s home position, and the central MRI frequency was calibrated. The alignment of the transducer’s focal point with the MRI system in all three axes was verified through short (10 s) low-power sonications (≤250 W). The HI-FU beam power was then gradually increased to achieve temperatures between 50 °C and 54 °C, producing transient clinical effects. Once the target was confirmed and no adverse effects were reported, the power was further increased (≥55 °C) to create a permanent lesion in the targeted volume. Following each cluster of sonications, intraoperative, high-resolution, T2-weighted images were acquired using a dedicated 2-channel head coil to visualize the resulting thalamic lesion. Additionally, a final high-resolution T2-weighted scan was always acquired at the end of the procedure to assess the lesion before discharging the patient from the FUS system. No steroid or osmotic therapies were routinely administered post-treatment; such therapies were only given if clinically indicated. Lesional longitudinal diameter was calculated on 48 h MRI follow-up T2-weighted images on an independent workstation using the length tool of the OsiriX MD software (build version 8 November 2024) in consensus by two experienced neuroradiologists [C.G and E.B. with 17 and 8 years of experience, respectively [24]. The lesional diameter included the maximum extension of the previously described characteristic lesional area of cytotoxic edema [25].

### 2.5. Statistical Analyses

Shapiro–Wilk’s test was carried out to check the normality of the quantitative variable. Continuous variables were reported as mean and standard deviation (SD) or by median and interquartile ranges (IQR) as appropriate. Categorical variables were reported as percentage and analyzed using Chi squared test or Fisher’s exact test accordingly. Repeated measures of continuous data were compared using paired Student’s *t*-test. The correlation between the FUS operating parameters and the quantitative study outcome were estimated by employing Spearman’s analyses with the calculation of Spearman’s correlation coefficient (rs). To explore factors associated with tcMRgFUS’s effectiveness, a time-to-event analysis was carried out by using a multivariable Cox-regression model. The dependent variable was a dichotomic event (partial/null-responder vs. fully responder), while the predictors were sex, age at tcMRgFUS, SDR ≥ 0.5, and baseline TETRAS PE ≥ 10. The time interval was the follow-up duration. If the event did not occur in this time interval, the case was defined as censored. The risk was expressed as hazard ratio (HR) and was reported with the 95% confidence interval (CI). For all the analyses, the level of statistical significance was set at a two-sided *p* value less than 0.5. The statistical analyses were performed using the Statistical Package for the Social Sciences software (IBM SPSS Statistics, Version 26.0; 2019. Armonk, NY, USA: IBM Corp).

## 3. Results

Sixty-one patients with ET underwent tcMRgFUS thalamotomy during the study period at the University Hospital Paolo Giaccone, Palermo, Italy. According to study inclusion criteria, *n* = 50 patients with ET were included in the final analyses, while the remaining ones were excluded due to incomplete follow-up data.

The majority of participants were right-handed (*n* = 49; 98%), men (*n* = 40; 80%), and they had a positive familiar history for ET (*n* = 36; 72%). At ET onset, the participants had a median age of 41.3 (20–57) years, while the median age at thalamotomy was 67.4 (58–76) years. Most patients presented with bilateral (*n* = 35; 70%) and segmental (*n* = 43; 86%) ET.

### 3.1. tcMRgFUS Procedure

All the right-handed patients underwent left tcMRgFUS thalamotomy, while the only left-handed patient underwent right thalamotomy.

The mean temperature achieved during lesioning sonications was 55.4 °C (range: 55–57 °C). The median number of lesioning sonications delivered was 2 (range: 1–3), resulting in a median lesion longitudinal diameter of 8.5 ± 2.4 mm. The median Skull Density Ratio (SDR) was 0.51 (range: 0.45–0.57). A complete overview of the tcMRgFUS operating parameters is provided in Table 1.

### 3.2. Impact of tcMRgFUS Thalamotomy on Tremor-Related, Self-Perceived Interference with Quality of Life

After tcMRgFUS thalamotomy, patients reported a slight increase in their global health status (70.9 ± 21.9 and 74.4 ± 21.4; +5% [95% CI: 0.9 to 10.9]), but this did not reach the statistical significance (*p* = 0.1; Figure 1A). Also, they reported a statistically significant improvement of self-perceived quality of life (70.2 ± 20.9 vs. 77.1 ± 18.4; +9.5% [95% CI: 3.1 to 16.4]; *p* = 0.005) and the reduction in the number of hours of tremor per day (19 ± 5.5 vs. 15 ± 6.2; −21% [95% CI: −31.4 to −10.5]; *p* < 0.0001), as shown in Figure 1A.

The total QUEST score decreased significantly after tcMRgFUS thalamotomy (32.9 ± 17.3 vs. 17.7 ± 9.8; −46.2% [95% CI: −62 to −30.4]; *p* < 0.0001), particularly in the hobbies (−36.7% [95% CI: −66.5 to −7.2]; *p* = 0.016), physical (−57.7% [95% CI: −69.1 to −46.1]; *p* < 0.0001, and psychosocial (−49.6% [95% CI: −68 to −31.1]; *p* < 0.0001) sub-items, as depicted in Figure 1B. However, tcMRgFUS thalamotomy did not affect the “Communication” and “Work” sub-items of the QUEST scale (Figure 1B).

### 3.3. Impact of tcMRgFUS Thalamotomy on Tremor-Related Disability in ADL and Tremor Severity

The total TETRAS-ADL subscore decreased significantly after tcMRgFUS thalamotomy (29.7 ± 6.8 vs. 14.2 ± 11.5; *p* < 0.0001) with a variation of −52.2% [95% CI: −62.9 to −41.4], as shown in Figure 2.

Similarly, while the total TETRAS-PE score was lower after thalamotomy in all participants (*p* < 0.0001; Figure 2), it was lower after tcMRgFUS thalamotomy on the treated side (11 ± 3.7 vs. 5.3 ± 3.7; *p* < 0.0001), reflecting a reduction of −51.8% [95% CI: −63 to −40.2]. As expected, the TETRAS-PE score on the untreated side was not affected by thalamotomy (*p* = 0.75; Figure 2).

Specifically, when analyzing each sub-item of the TETRAS-ADL scale, we found that the self-perceived disability due to ET reduced in all the TETRAS-ADL sub-items after tcMRgFUS thalamotomy, except for speaking task (−14.3% [−14.3 to 48.6], as reported in Table 2. The highest variation was found in the use of spoon (−70% [−80 to −52]) and drinking (−62.1% [−75.9 to −49.3]) tasks as well as in the use of keys (−61.6% [−76 to −48], as reported in Table 2.

### 3.4. Correlation Between tcMRgFUS Operating Parameters and Clinical Outcomes

Furthermore, we analyze the linear relationship between tcMRgFUS-operating parameters, QUEST total score, and TETRAS-PE. The temperature peak (°C) of delivered sonication and the amount of QUEST Total score reduction (rs = 0.38, *p* = 0.0073) were higher (Figure 3A). Similarly, the amount of QUEST Total score reduction was positively correlated with the number of delivered sonications (rs = 0.42, *p* = 0.005; Figure 3B) and the longitudinal lesion diameter (rs = 0.30, *p* = 0.030; Figure 3C).

Finally, the degree of TETRAS-PE reduction was positively correlated with the longitudinal lesion diameter (rs = 0.3, *p* = 0.02; Figure 3D). No other statistically significant correlation emerged between the FUS operating parameters (i.e., SDR, energy peak, average temperature, median delivered energy) and the TETRAS-ADL, TETRAS-PE, and QUEST scores.

### 3.5. Predictors of tcMRgFUS Thalamotomy Effectiveness

By considering the TETRAS-PE variation, 27 patients (54%) reported at least a 50% reduction in this outcome after tcMRgFUS (mean variation −75% [95% CI: −81.7 to −68.9]), while the remaining participants (46%) showed a less than 50% variation in TETRAS-PE (mean variation −19% [95% CI: −31.5 to −6.6]). After performing a multivariable Cox regression analysis, a baseline TETRAS-PE score ≥ 10 was singled out as a predictor of MRgFUS’s effectiveness (HR 6.6 [95% CI: 2.1–21.3]; *p* = 0.001), while age at MRgFUS (HR = 1.8 [95% CI 0.7–4.2], *p* = 0.2), SDR ≥ 0.5 (HR = 1.5 [95% CI 0.7–3.5], *p* = 0.3), and male sex (HR = 0.97 [95% CI 0.3–3], *p* = 0.9) were not statistically significant or associated with the outcome.

### 3.6. Safety Analyses

A total of 36 subjects (72%) experienced at least one MRI- or sonication-related AE, while *n* = 12 (24%) participants presented with thalamotomy-related AE (Table 3).

The most common MRI- or sonication-related AE was pin site pain (58%), while the most common thalamotomy-related AEs were paresthesia or limb numbness (10%) and balance disorder (10%). The relative frequencies of the MRI- and thalamotomy-related AE are reported in Table 3. The frequencies of both MRI and thalamotomy AEs were comparable between the male and female participants (*p* > 0.05). The occurrence of a thalamotomy-related AE was positively correlated with lesion longitudinal diameter (rs = 0.3, *p* = 0.035) and SDR (rs = 0.29, *p* = 0.045), but no other associations emerged. Out of the 12 patients reporting tcMRgFUS thalamotomy-related AEs, *n* = 1 (8.3%) reported persistent tongue paresthesia at the last 9-month follow-up, while AEs were transient in other cases. Finally, most of the participants (*n* = 41; 82%) expressed the willingness to repeat the FUS thalamotomy, while *n* = 9 subjects (18%) declared that they would never do it again.

## 4. Discussion

In this real-world, prospective cohort study, we explored the effectiveness and safety of unilateral tcMRgFUS thalamotomy in fifty patients with ET by evaluating self-reported and objective outcome measures. After tcMRgFUS thalamotomy, the patients reported a significant improvement in the quantitative, self-reported assessment of quality of life (+9.5%) as well as a significant reduction in the number of hours of tremor per day (−21%); however, the global health status was not affected by thalamotomy (Figure 1A). All the participants underwent the QUEST scale before and after thalamotomy. They reported a significant reduction in the QUEST total score after tcMRgFUS thalamotomy (−46.2%), and this reduction was marked in the hobbies (−36.7%), physical (−57.7%), and psychosocial domains (−49.6%), as depicted in Figure 1B. This means that the self-perceived, tremor-related interference in the explored quality-of-life domains dropped after tcMRgFUS thalamotomy, especially in the hobbies, physical, and psychosocial domains; communication and work were less affected by tcMRgFUS thalamotomy, probably due to less interference with tremor in these domains at baseline (Figure 1A). Our results are in line with existing ones, confirming the positive effect of tcMRgFUS thalamotomy on the quality of life of patients with ET [26,27]. When evaluating the grade of disability in ADL, we found a decrease in TETRAS-ADL score of up to 52.2% after thalamotomy, suggesting a global reduction in disability in ADL tasks. Interestingly, the highest reduction in disability concerned tasks requiring proper hand dexterity, such as drinking, use of spoon, use of keys, and writing (Table 2). It is worth noting that tcMRgFUS thalamotomy did not affect the disability level of speaking; obviously, in absence of voice tremor, it is reasonable that speaking tasks remain unaffected. Furthermore, concerning the objective outcome measures, we found a reduction in total TETRAS-PE score (Figure 2), which was driven by the reduction in TETRAS-PE score on the treated side by tcMRgFUS thalamotomy, as expected (Figure 2). Thus, the participants reported an improvement in their quality of life and a reduced disability in ADL task after thalamotomy. In addition, we documented a reduction in tremor severity after thalamotomy by performing TETRAS-PE. Until now, the most used clinical outcomes in previous tcMRgFUS thalamotomy studies were QUEST and Clinical Rating Scale for Tremor (CSRT). In this study, for the first time, we employed TETRAS-ADL and TETRAS-PE to evaluate the effectiveness of tcMRgFUS thalamotomy on ADL disability and tremor severity in patients with ET, respectively. As a strength, TETRAS-ADL may allow us to explore multiple disability domains compared to QUEST despite maintaining a good correlation [28]. On the other hand, TETRAS-PE may reveal some advantages over CRST in the assessment of tremor severity, such as being also correlated with CRST [29]. Thus, our data are in line with those reported by other studies that documented a significant reduction in tremor severity evaluated through the CRST score one year after tcMRgFUS thalamotomy [26,27,30]. In this view, our results further highlight the positive impact of tcMRgFUS thalamotomy in tremor-related interference of quality of life, disability in ADL, and tremor severity in patients with ET. In addition, we explored the linear correlation between tcMRgFUS-operating parameters and study outcomes. We found that the magnitude of QUEST score reduction was higher than the operating temperature peak, and the number of delivered sonications and lesion longitudinal diameter was also greater. These results mean that the right setting of procedural parameters may result in better clinical outcomes. It is worth noting that we did not find any correlation between SDR and the study outcomes. However, patients with an SDR below 0.3 were not eligible for tcMRgFUS, resulting in a lack of linear correlation between SDR and clinical outcomes, since patients with lower probability of improvement due to lower SDR were a priori excluded. These data are in line with those reported in previous studies [31,32]. Moreover, we investigated whether such putative factors were associated with complete tcMRgFUS effectiveness. Specifically, patients were considered as full responders to tcMRgFUS when a reduction of at least 50% was documented in the TETRAS-PE score. This was observed in 54% of the study participants; younger age at tcMRgFUS thalamotomy and higher SDR were associated with better outcome. Although this result was not statistically significant, a higher baseline tremor severity was the strongest predictor of tcMRgFUS thalamotomy’s success. Our data are partly discordant to existing findings. Torii et al. reported that lower baseline tremor disability was a positive prognostic factor 1 and 3 months after tcMRgFUS [31]. Although we did not perform the 1-, 3- and 6-month evaluations, our results, combined with those of Torii et al., may indicate an immediate beneficial effect of tcMRgFUS in patients with lower tremor disability but a longer-lasting effect in those with higher baseline tremor severity. Finally, 72% and 24% of participants reported at least one MRI- or ultrasonographic- and thalamotomy-related AEs, respectively. The most common MRI-related AE was pin site pain, while the most common thalamotomy-related AEs were balance disorder and limb numbness. We found that the number of AEs was positively related to the longitudinal lesion diameter and SDR, according to the existing data [31]. Only one study subject reported a permanent AE, resulting in a more favorable safety profile of tcMRgFUS thalamotomy in our study compared to others [26,27,30,33]. Thus, in the past decade, tcMRgFUS has achieved increasing recognition at both international and national levels as an effective and safe treatment for ET [10,34,35,36]. The relatively recent approvals of staged bilateral (9 months) tcMRgFUS treatments for ET by both the FDA (January 2023) and the CE (September 2023) are expected to further support the accumulation of clinical evidence and broaden its adoption in clinical practice [37,38].

## 5. Strengths and Limitations

This study has some limitations. First, the low number of participants might have affected the results of multivariable regression analyses, as shown by the non-statistically significant results and wide CIs. Moreover, we did not include FUS-operating parameters in the regression analyses. Thus, these results should be interpreted carefully. Another limitation is the short follow-up time of one year. However, as the strengths of our study, we highlight its real-world design, the inclusion of a higher number of participants compared to other similar studies, as well as the concurrent use of QUEST and TETRAS to evaluate tremor severity.

## 6. Conclusions

This real-world cohort, single-center study confirms the efficacy and safety of unilateral VIM tcMRgFUS thalamotomy in patients with drug-refractory ET, highlighting a significant improvement in tremor-related interference in quality-of-life domains, ADL tasks, and tremor severity. Notably, in our study, TETRAS-ADL and TETRAS-PE, for the first time, have been concurrently employed to evaluate functional disability and tremor severity, respectively, offering a more comprehensive clinical picture compared to previous studies. The growing recognition by both scientific societies and regulatory authorities is expected to further consolidate the evidence base, expand clinical indications, and enhance accessibility to this minimally invasive therapeutic option. Taken together, our findings further solidify the role of tcMRgFUS in the therapeutic algorithm of ET, offering measurable benefits in tremor control, quality of life, and functional independence while maintaining a favorable safety profile. Future studies with larger sample sizes and longer follow-up periods, particularly in the context of bilateral treatments, will be crucial to confirm and expand upon these results. We advocate for increased attention to both patient self-reported and objective outcome measures data in future studies, as these measures provide essential insights into the real-life impact of treatments on quality of life and daily functioning.

## Figures and Tables

**Figure 1 neurolint-17-00131-f001:**
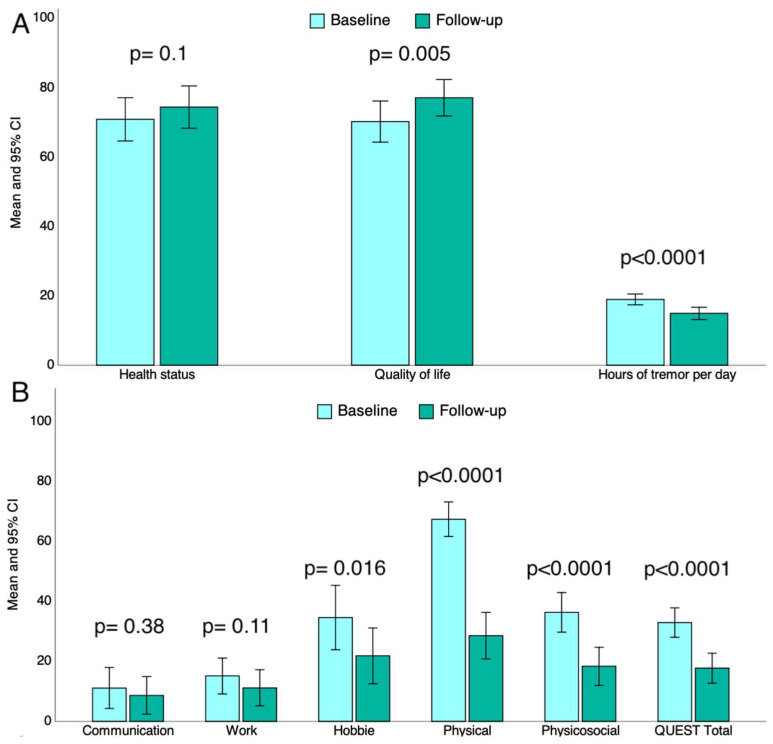
Bar charts showing the comparison between the baseline and follow-up times after tcMRgFUS of the self-reported quality of life (from 0 to 100), health status (from 0 to 100), and number of hours of tremor per day (from 0 to 24) in panel (**A**) and the total QUEST score and its sub-items in panel (**B**). Means and 95% CI are reported on the y axis. CI, confidence interval; QUEST, Quality of Life in Essential Tremor Questionnaire; tcMRgFUS, transcranial magnetic resonance-guided focused ultrasound.

**Figure 2 neurolint-17-00131-f002:**
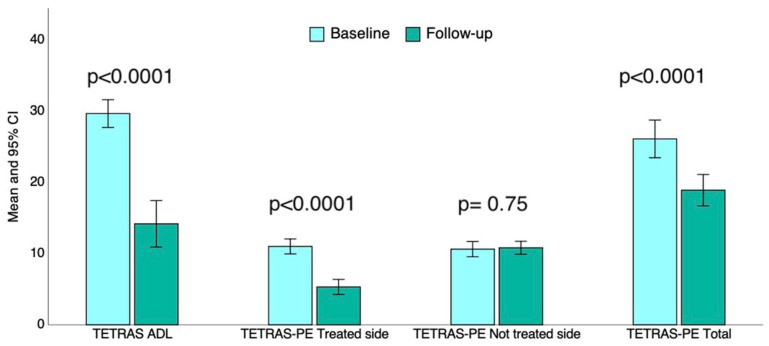
Bar charts showing the comparison between baseline and follow-up time after tcMRgFUS of the TETRAS-ADL subscore, total TETRAS-PE, and TETRAS-PE on the treated and untreated sides, respectively. TETRAS, TRG Essential Tremor Rating Assessment Scale; ADL, activity daily living; PE, performance; tcMRgFUS, transcranial magnetic resonance-guided focused ultrasound.

**Figure 3 neurolint-17-00131-f003:**
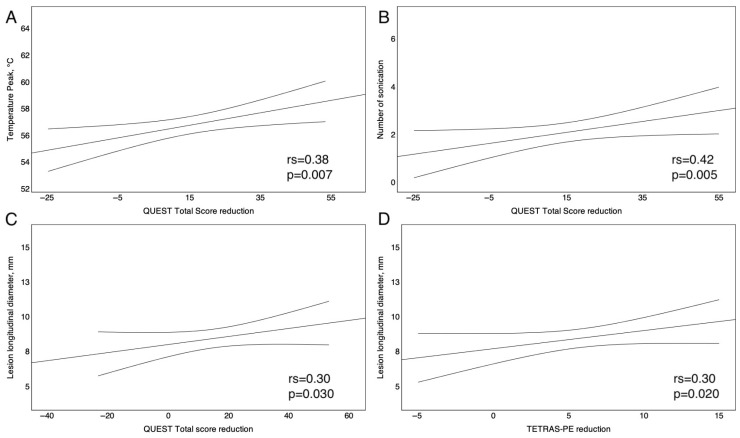
Line graphs showing the linear correlations between the amount of QUEST total score reduction and the temperature peak of delivered sonications (**A**), the number of delivered sonications (**B**), the lesion longitudinal diameter (**C**), and the linear correlation between the degree of TETRAS-PE reduction and the lesion longitudinal diameter (**D**). QUEST, Quality of Life in Essential Tremor Questionnaire; TETRAS, TRG Essential Tremor Rating Assessment Scale; PE, performance; rs, Spearman’s r. The upper and lower curved lines represent the 95% confidence intervals.

**Table 1 neurolint-17-00131-t001:** tcMRgFUS operating parameters.

Parameters	
Average temperature °C of lesioning sonications, median (IQR)	55.4 (55–57)
Temperature peak °C, median (min and max)	57 (53–64.4)
Median delivered energy J, median (IQR)	13,821 (10,116–25,274)
Energy peak J, median (min and max)	14,306 (7989–94,694)
Number of lesional sonication delivered, median (IQR)	2 (1–3)
Skull density ratio, median (IQR)	0.51 (0.45–0.57)
Lesion longitudinal diameter at 48 h MRI, mean (SD)	8.5 (2.4)

MRgFUS, magnetic resonance-guided focused ultrasound; ET, essential tremor; IQR, interquartile range; SD, standard deviation; MRI, magnetic resonance imaging.

**Table 2 neurolint-17-00131-t002:** Comparison of the TETRAS-ADL sub-items score before and after tcMRgFUS thalamotomy with percentual variation.

TETRAS-ADL Sub-Items	Baseline Mean (SD)	Follow-Up Mean (SD)	Mean Variation% (95%CI)		*p*
Speaking	0.7 (1.02)	0.6 (0.84)	−14.3	(−14.3 to 48.6)	0.28
Spoon	3 (0.9)	0.9 (1.29)	−70.0	(−80 to −52)	<0.0001
Drinking	2.9 (0.87)	1.1 (1.28)	−62.1	(−75.9 to 49.3)	<0.0001
Hygiene	1.7 (1.4)	0.8 (1.17)	−52.9	(−74.7 to 29.4)	<0.0001
Dressing	1.6 (1.1)	0.9 (1.11)	−43.8	(−62.5 to −23.1)	<0.0001
Pouring	2.8 (1.02)	1.4 (1.33)	−50.0	(−64.3 to −37.5)	<0.0001
Trays	3.2 (0.9)	1.9 (1.48)	−40.6	(−53.8 to −28.1)	<0.0001
Keys	2.5 (1.04)	0.96 (1.21)	−61.6	(−76 to −48)	<0.0001
Writing	3.2 (0.86)	1.4 (1.24)	−56.3	(−68.8 to −45.3)	<0.0001
Working	2.6 (0.92)	1.3 (1.24)	−50.0	(−68.8 to −37.3)	<0.0001
Social impact	2 (1.51)	1.1 (1.3)	−45.0	(−70 to −31.5)	<0.0001

TETRAS, TRG Essential Tremor Rating Assessment Scale; SD, standard deviation; CIs, confidence intervals; ADL, activity daily living; tcMRgFUS, transcranial magnetic resonance-guided focused ultrasound.

**Table 3 neurolint-17-00131-t003:** Relative frequencies of the MRI-, sonication-, and related thalamotomy-related adverse events reported by the participants in this study.

Adverse Event	Frequency, *n* (%)
MRI- and sonication-related	
Pin site pain	29 (58)
Cold sensation (due to degassed and cooled water)	22 (44)
Headache (during high-power sonications)	11 (22)
Neck, back, or shoulder pain	10 (20)
Thalamotomy-related *	
Limb or facial paresthesia/numbness	5 (10)
Hemiparesis	3 (6)
Balance disorder	5 (10)
Dysarthria	2 (4)

MRI, magnetic resonance imaging. * Observed as transient symptoms in all patients except one that reported persistent tongue paresthesia at the last clinical follow-up.

## Data Availability

The data are available upon specific request to corresponding author.

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
