# Peer review of "Safety and Effectiveness of Unilateral Transcranial Magnetic Resonance-Guided Focused Ultrasound in Essential Tremor: One-Year Single-Center Real-World Results"

_2035-8377, 2025, doi:10.3390/neurolint17080131_

Round 1
Reviewer 1 Report
Comments and Suggestions for Authors
This work introduces the safety and effectiveness of TcMRgFUS thalamotomy in in patients with essential tremor based on a one-year single-center clinical results.
1) What’s about the content of QUEST Questionnaire? What’s the representative meaning of the QUEST results for the safety and effectiveness evaluation of TcMRgFUS? Why the QUEST index decreases after TcMRgFUS procedure? Is it means the quality life decrease after treatment?
2) There are some evaluation methods in the work. The representative meaning of these methods are suggested to explain clearly. It is better for the reader to understand.
3) There're some errors in the Figure 2 (or figure 3?) labels in the text. The letters in Figure 2 is too small and letter sizes are not uniform.
4) What’s the meaning and assignment of the three curves in Figure 2. And they are not scatterplots as noted in the caption.
Author Response
We thank the reviewer for his comments.
Comment 1)What’s about the content of QUEST Questionnaire? What’s the representative meaning of the QUEST results for the safety and effectiveness evaluation of TcMRgFUS? Why the QUEST index decreases after TcMRgFUS procedure? Is it means the quality life decrease after treatment?
R: We thank the reviewer for the careful reading of our manuscript. We forgot to describe QUEST scale in the methods section. As QUEST scale has been devleoped to explore the grade of interference of ET with perceived quality of life, higher score indicates higher perceived interference. This explain why QUEST score was reduced after FUS in our cohort indicating a reduced perceived tremor-related interference by patients. Conversely, the variable “quality of life” analyzed in Figure 1 is a self-reported visual analogic scale going from 0 to 100 as its higher values indicate a better quality of life. To avoid confusion between quality of life evaluated through VAS scale and QUEST score, we revised the methods section accordingly as follows: “QUEST score is a 30-item composite scale developed to measure the impact of ET in quality of life by exploring the interference of tremor in five domains (i.e., Communication, Work and Finance, Hobbies and Leisure, Physical, Psychosocial) (17). The grade of interference of ET in each domain is explored by providing specific questions which are scored from 0 (no interference) to 4 (always). The sum of each subscore turns in the total QUEST score, thus, higher values indicate higher interference of EF with patient perceived quality of life (17)”. Also we revised the reference list accordingly.
Comment 2) There are some evaluation methods in the work. The representative meaning of these methods are suggested to explain clearly. It is better for the reader to understand.
R: According to your suggestions. Methods section has been clarified. See the changes in red.
Comment 3) There're some errors in the Figure 2 (or figure 3?) labels in the text. The letters in Figure 2 is too small and letter sizes are not uniform.
R: Figure 2 has been totally revised accordingly. Since we divided the previous Figure 1 in new Figure 1 and Figure 2, due to other comments, the previous modified Figure 2 is now called Figure 3.
Comment 4) What’s the meaning and assignment of the three curves in Figure 2. And they are not scatterplots as noted in the caption.
R: In the new 3.4 results section “Correlation between tcMRgFUS operating parameters and clinical outcomes” we explored the linear correlation between FUS operating parameters and clinical outcomes (reduction of QUEST and TETRAS-PE) exploring whether temperature peak, average temperatur, SDR, energy peak, median delivered energy, number of sonication and longidutinal diameters were correlated with the reduction of TETRAS and QUEST. We found that number of sonication and longitudinal lesion diameter were directly correlated with TETRAS and QUEST reduction meaning an increased quality of life and reduced tremor-related disability, respectively. No correlation emerged between SDR and other parameters with clinical outcomes, thus these line charts were not reported. We reviewed the 3.2 results section as appropriate (see the changes in red). In our opinion these analyses should be included in the paper showing to the reader the importance of a correct setting of operating parameters (as other authors did, see the ref. 31-32). Moreover, we found that lesion diameters were correlated with the occurrence of Adverse events, making furthermore relevant these analyses. We agree with the reviewer, new Figure 3 does not represent scatterplot, we changed Line graphs instead scatterplot as appropriate and revised the Figure legend as follows: “Figure 3. Line graphs showing the linear correlations between the amount of QUEST total score reduction and temperature peak of delivered sonications (A), number of delivered sonications (B) and lesion longitudinal diameter (C), and the linear correlation between the amount of TETRAS-PE reduction and lesion longidutinal diameter (D). QUEST, Quality of Life in Essential Tremor Questionnaire; TETRAS, TRG Essential Tremor Rating Assessment Scale; PE, performance; rs, Spearman’s r. The upper and lower curved lines represent the 95% confidence intervals.”
Reviewer 2 Report
Comments and Suggestions for Authors
Please check comments to authors:
It is an interesting manuscript about analysis Essential Tremor (ET) illness by Unilateral transcranial Magnetic Resonance guided Big Data sources within one year period of time.
It merits to be published; however please consider the following comments and suggestions:
i) in the title, please consider to avoid abbreviations.
ii) it is suggested to revise English writing; such as in the statement of aims.
iii) About introduction and the aims of the research. It is not clear the source of data. Is it a Big Data? Few cases? Please revise the redaction and explanations for better understanding.
iv) About materials and methods; please change of place the aims section incorporated. It is not the right the actual place in the methods.
v) in the methods; it should be mentioned how the samples were chosen?and more details should be addressed such as age of patients, health history, etc.
vi) please revise the need of literature of methods; and highlight if methods were developed by authors or modified from previous reports.
vii) in the section of results; please revise the subsection assignations. It seems that should be added one unless.
viii) in the introduction of the section of results it is not well described the table. And there are results into the table as parameters that should be described to then analyze them, discuss, and conclude.
ix) please check the need to add more legends in table 2 for better interpretation in absence of the text.
x) section 3. 2 of results is too short. It should be described and analyzed to then leave to the discussion.
xi) about the conclusions. They show interesting perspectives and even additional variables with references were added. But, it is just a suggestion; please revise it considering the deep discussion developed.
Comments on the Quality of English LanguagePlease check comments to authors.
Author Response
Comment 1) in the title, please consider to avoid abbreviations.
R: we agree with your comments. We changed the title as appropriate “Safety and effectiveness of unilateral transcranial magnetic resonance-guided focused ultrasound in essential tremor: One-Year Single-Center Real-World Results”.
Comment 2) it is suggested to revise English writing; such as in the statement of aims.
R: the manuscript underwent extended language revision. See the changes in red.
Comment 3) About introduction and the aims of the research. It is not clear the source of data. Is it a Big Data? Few cases? Please revise the redaction and explanations for better understanding.
R: In the introduction and aims section we clarified the study type accordingly:”In this prospective, real-word, cohort-study we explored the safety and effectiveness of tcMRgFUS thalamotomy in patients with drug refractory ET attending to Movement disorder Outpatient service of University-Hospital Paolo Giaccone, Palermo, Italy”.
The source of data are explained in “Study design and participants “section as follows: “A prospective cohort study was carried out by enrolling patients with ET who underwent tcMRgFUS thalamotomy at the University Hospital Policlinico Paolo Giaccone, Palermo”. The number of enrolled participants and those who were excluded is reported in the introduction of results section. Accordingly, this section has been clarified as follows: ” Sixty-one patients with ET underwent tcMRgFUS thalamotomy during the study period at the University-Hospital Paolo Giaccone, Palermo, Italy. According to study inclusion criteria, n= 50 patients with ET were included in the final analyses while the remaining ones were excluded due to incomplete follow-up data”.
Comment 4) About materials and methods; please change of place the aims section incorporated. It is not the right the actual place in the methods.
R: We reviewed the methods section accordingly and we moved the “Objective” paragraph at the end of introduction as follows: “In this prospective, real-word, cohort-study we explored the safety and effectiveness of tcMRgFUS thalamotomy in patients with drug refractory ET attending to Movement disorder Outpatient service of University-Hospital Paolo Giaccone, Palermo, Italy. Specifically, aims of this study were to: 1) explore the impact of unilateral tcMRgFUS thalamotomy in the perceived tremor-related interference in quality of life and trem-or-related ADL disability in a cohort of patients with drug-refractory ET; 2) explore factors associated with tcMRgFUS thalamotomy effectiveness; 3) explore the relationship be-tween tcMRgFUS operating parameters, tremor-related quality of life and tremor-related disability; 4) assess the safety of unilateral tcMRgFUS thalamotomy”.
Comment 5) in the methods; it should be mentioned how the samples were chosen? and more details should be addressed such as age of patients, health history, etc.
R: In the methods section “Study design and participants” we reported the study inclusion criteria as follows: “The study inclusion criteria were to have: 1) a diagnosis of ET according to the Consensus Statement on the Classification of tremors (14); 2) a drug-refractory ET; 3) a skull deviation ratio (SDR) ≥ 0.3; 4) complete clinical and tcMRgFUS data; 5) a follow-up of at least one year.” No other criteria (age, history, etc) were chosen for inclusion. We added this sentence in the” Study design and participants” section: “No other study inclusion criteria such as age or medical history were chosen.” Moreover, we clarified in the results section how participants were excluded as follows “Sixty-one patients with ET underwent tcMRgFUS thalamotomy during the study period at the University-Hospital Paolo Giaccone, Palermo, Italy. According to study inclusion criteria, n= 50 patients with ET were included in the final analyses while the remaining ones were excluded due to incomplete follow-up.”
Comment 6) please revise the need of literature of methods; and highlight if methods were developed by authors or modified from previous reports.
R: The Methods section has also been revised in response to comments from another reviewer. We believe that, in its current form, it is technically more robust and scientifically more coherent and readable. Regarding the methodology of the TcMRgFUS thalamotomy procedure under investigation, the relevant literature supporting the operative protocol adopted at our center has already been cited in the manuscript (Refs. 19–22), along with the tractography-based targeting approach (Refs. 23–24), the lesional areas considered for lesion diameter assessment (Ref. 26), and the software used for these measurements (Ref. 25). That said, should the reviewer wish to further clarify or expand on this request, the authors remain fully available to consider additional modifications.
Comment 7) in the section of results; please revise the subsection assignations. It seems that should be added one unless.
R: according your suggestion, we added the subsection “3.1. tcMRgFUS procedure” discussing also Table 1, as follows “All the right-handed patients underwent left tcMRgFUS thalamotomy while the only one left-handed patient underwent right thalamotomy. The mean temperature of lesioning sonications was 55.4°C (55-57) and the median number of delivered sonication was 2 (1-3) obtaining a median lesion longitudinal di-ameter of 8.5mm ± 2.4. The median SDR was 0.51 (0.45-0.57). The complete tcMRgFUS operating parameters are reported in Table 1”. Moreover we divided the previous section 3.2. in 3.2 “Impact of MRgFUS thalamotomy on tremor-related, self-perceived, interference in quality of life” describing the impact of MRgFUS in self perceived interference due to ET evaluated by QUEST and 3.3 “Impact of tcMRgFUS thalamotomy on tremor-related disability in ADL and tremor severity” wherein we discussed the variation of TETRAS-ADL subitems and TETRAS-PE. Also, we divided the previous Figure 1 in Figure 1 (QUEST) and Figure 2 (TETRAS-ADL and TETRAS-PE), placing them in the appropriate sections (new Figure 1 in section 3.2 and new Figure 2 in section 3.3). We also expanded the results section, accordingly.
Comment 8 in the introduction of the section of results it is not well described the table. And there are results into the table as parameters that should be described to then analyze them, discuss, and conclude.
R: we agree with the reviewer comments. We avoided to describe parameters in the text since we reported them in the Table 1. However, We added a new subsection 3.1 tcMRgFUS procedure as follows: “All the right-handed patients underwent left tcMRgFUS thalamotomy while the only one left-handed patient underwent right thalamotomy. The mean temperature of lesioning sonications was 55.4°C (55-57) and the median number of delivered sonication was 2 (1-3) obtaining a median lesion longitudinal di-ameter of 8.5mm ± 2.4. The median SDR was 0.51 (0.45-0.57). The complete tcMRgFUS operating parameters are reported in Table 1”. Also the new sections 3.2 and 3.3 were expanded presenting results not discussed before.
Comment 9) please check the need to add more legends in table 2 for better interpretation in absence of the text.
R: We enriched the legends. According to this and previous reviewer’s suggestions we divided the previous Figure 1 in Figure 1 (QUEST) and Figure 2 (TETRAS-ADL and TETRAS-PE) discussing these results separately Improving the clarity of the results' presentation for the reader. We also removed redundant data from Table 2. Table 2 is now discussed throughout the text as follows: “Specifically, when analyzing each sub-items of TETRAS-ADL scale, we found that the self-perceived disability due to ET reduced in all the TETRAS-ADL sub-items after tcMRgFUS thalamotomy, at the exception of speaking task (-14.3% [-14.3 to 48.6] as re-ported in Table 2. Specifically, the highest variation was found in the spoon (-70% [-80 to -52]) and drinking (-62.1% [-75.9 to -49.3]) tasks as well as in the use of keys (-61.6% [-76 to -48] as reported in Table 2”.
Comment 10) section 3. 2 of results is too short. It should be described and analyzed to then leave to the discussion.
R: We enlarged the section accordinly as follows: “Furthermore, we analyze the linear relationship between tcMRgFUS operating parameters, QUEST total score and TETRAS-PE. Higher was temperature peak °C of delivered sonication and higher was the amount of QUEST Total score reduction (rs=0.38, p=0.0073; Figure 3A). At the same manner, the amount of QUEST Total score reduction was positively correlated with number of delivered sonication (rs=0.42, p=0.005; Figure 3B) and longitudinal lesion diameter (rs=0.30, p=0.030; Figure 3C). Finally, the amount of TETRAS-PE reduction was positively correlated with longitudinal lesion diameter (rs=0.3, p=0.02; Figure 3D). No other statistically significant correlation emerged between FUS operating parameters (i.e., SDR, energy peak, average tem-perature, median delivered energy), TETRAS-ADL, TETRAS-PE and QUEST score”. We also better clarified the legend of Figure 3. Moreover, we further discussed the results reported in the 3.4 section as follows: “In addition, we explored the linear correlation between tcMRgFUS operating parameters and study outcomes. We found that the magnitude of QUEST score reduction was higher as the operating temperature peak, the number of delivered sonications and lesion longidutinal diameter were greater. These results would mean that the right setting of procedural parameters may turn in better clinical outcomes. It is worth nothing that we did not find any correlation between SDR and study outcomes. However, patients with an SDR below 0.3 did not eligible to tcMRgFUS turning in a lack of linear correlation be-tween SDR and clinical outcomes since patients with lower probability of improvement due to lower SDR were a priori excluded. These data are in line to that reported in previous studies (31-33).”
Comment 11) about the conclusions. They show interesting perspectives and even additional variables with references were added. But, it is just a suggestion; please revise it considering the deep discussion developed.
R: we agree with reviewers comment and we shortened the conclusion section by moving additional variables and their references at the bottom of discussion section. We created a new paragraph “5. Strengths and Limitations” section, clarifying the discussion section.
Round 2
Reviewer 2 Report
Comments and Suggestions for Authors
Thank you very much for your responses and modifications applied. Now, from the Reviewer point of view the manuscript was improved, and it merits to be published. So, it is accepted for publication.